# Non-Contact Detection of Vital Signs Based on Improved Adaptive EEMD Algorithm (July 2022)

**DOI:** 10.3390/s22176423

**Published:** 2022-08-25

**Authors:** Didi Xu, Weihua Yu, Changjiang Deng, Zhongxia Simon He

**Affiliations:** 1School of Integrated Circuits and Electronics, Beijing Institute of Technology, Beijing 100081, China; 2Microwave Electronics Laboratory, Department of Microtechnology and Nanoscience, Chalmers University of Technology, 41296 Gothenburg, Sweden

**Keywords:** non-contact vital signs detection, ensemble empirical mode decomposition (EEMD), frequency modulated continuous wave (FMCW), static clutter filtering

## Abstract

Non-contact vital sign detection technology has brought a more comfortable experience to the detection process of human respiratory and heartbeat signals. Ensemble empirical mode decomposition (EEMD) is a noise-assisted adaptive data analysis method which can be used to decompose the echo data of frequency modulated continuous wave (FMCW) radar and extract the heartbeat and respiratory signals. The key of EEMD is to add Gaussian white noise into the signal to overcome the mode aliasing problem caused by original empirical mode decomposition (EMD). Based on the characteristics of clutter and noise distribution in public places, this paper proposed a static clutter filtering method for eliminating ambient clutter and an improved EEMD method based on stable alpha noise distribution. The symmetrical alpha stable distribution is used to replace Gaussian distribution, and the improved EEMD is used for the separation of respiratory and heartbeat signals. The experimental results show that the static clutter filtering technology can effectively filter the surrounding static clutter and highlight the periodic moving targets. Within the detection range of 0.5 m~2.5 m, the improved EEMD method can better distinguish the heartbeat, respiration, and their harmonics, and accurately estimate the heart rate.

## 1. Introduction

The remote monitoring of human vital signs has aroused great interest of researchers in various fields, such as medical monitoring, anti-terrorism action, as well as rescue action and security. Vital signs mainly include heart rate, respiratory rate. It can be used to assist doctors in treatment, daily family health monitoring, morbid respiratory pattern monitoring and sleep quality evaluation [1,2,3]. It is not easily affected by temperature, humidity, working environment and other factors, which not only provides a non-invasive and convenient means of detecting vital signs, but also makes special scene applications possible. For example, vital sign information detection is carried out for critical patients such as large-area burn and trauma, infectious patients, and mental patients [4,5], in rescue, counter-terrorism response, emergency searches [6,7], in monitoring of infants and detection of sleep disorders in adults [8].

At present, there are three main types of radar systems used for vital signs detection, namely pulse radar (UWB radar) [9], CW doppler radar [10,11,12], and frequency modulated continuous wave (FMCW) radar [7,13,14,15,16,17,18]. CW doppler radar is not good at distinguishing clutter from multiple targets because of the lack of modulation spectrum information. Therefore, the vital sign monitoring method based on CW Doppler radar only relies on Doppler information to detect relative motion [1,2,19,20,21]. Moreover, the limitations of CW in detecting distance can be a disadvantage of using the system [22]; the extracted Doppler information may contain micro-motion interference from the human body and the nearby environment, resulting in low detection accuracy of vital signs. Compared with CW radar and pulse radar, FMCW radar has higher range resolution and velocity resolution, it can distinguish multiple targets [23,24], and extract target micromotion information, and it is the mainstream choice in the field of vital signs detection [1,6,7]. Therefore, this paper adopts FMCW radar for vital sign detection. In addition, because non-contact vital signs detection mainly depends on the displacement of the chest caused by breathing heartbeat detection, and heart rate changes caused by the tiny displacement can cause significant change to the phase, the low resolution of low-frequency radar may lead to inaccurate detection results. Some studies on FMCW systems with several millimeter-wave frequencies such as 24 GHz [23,25], 77 GHz [15,26], and 220 GHz [27] have been carried out to detect non-contact human breathing. Actual application of millimeter wave technology greatly improves the detection accuracy, and makes the non-contact detection of vital signs possible in practical applications. Table 1 lists the different methods and performance of frequency modulated Continuous wave (FMCW) radar for vital sign monitoring.

Alizadeh et al. [15] applied 77 GHz millimeter-wave radar to detect the components of vital signals by extracting the phase of intermediate frequency (IF) signals. The correlation between the respiratory and heartbeat rates estimated by the reference sensor and radar is 94% and 80%, respectively. The proposed MTI-autocorrelation-EEMD can reconstruct the respiratory and heartbeat signals well, but the detection range applied in the experiment is short [9]. Xu Zhengwu et al. [27] studied the detection of human vital characteristics based on 220 GHz solid-state source terahertz biologic radar system, using Empirical Mode Decomposition. However, as mentioned in [28,29], EMD has problems of modal mixing and endpoint effect, which cannot separate respiratory and heartbeat signals well, which limits its application in practice. A joint spectral estimation method based on adaptive recognition embedded Ensemble Empirical Mode Decomposition (EEMD) is proposed for heart rate measurement [30]. Experimental results show that in the detection range of 1–2.5 m, the method can better detect distracted jumping and breathing, and the root mean square error is less than 6 BPM.

Several issues need to be considered for vital sign detection based on the EEMD method, such as (i) the need for a high-resolution feature to detect small displacements on the chest or abdominal wall. Conventional viewpoints in radar systems require a wide range of bandwidth signals. (ii) In the traditional EEMD algorithm, there are respiratory harmonics close to the heartbeat frequency in the decomposed heartbeat component, which hinders the correct assessment of heart rate. (iii) The clutter contributed by the non-periodic moving obstacles around the target, especially those close to the target, will make the system output wrong phase information. A way is needed to overcome the aforementioned problems. Thus, the robustness and accuracy of FMCW radar system in respiratory vital sign detection application can be enhanced. The problem to be solved in this paper is to suppress aperiodic motion information, overcome the influence of respiratory harmonics, and improve detection efficiency.

In this paper, an integrated empirical mode decomposition (EEMD) algorithm based on noise distribution features and a static clutter filtering method are proposed. The static clutter filtering method filters the information of static and aperiodic moving objects in the environment. Then, the phase accumulation method is used to enhance the robustness of phase extraction. In addition, the DACM algorithm is used to identify the phase signal of each chirp to solve the phase jump problem. Finally, the improved EEMD eliminates the influence of respiratory harmonics and improves the detection efficiency. This paper is organized as follows: Section 2 explains the principle of FMCW radar vital sign measurement. In Section 3, the implementation steps of the proposed algorithm are introduced. The experimental results and conclusions are given in Section 4 and Section 5, respectively.

## 2. Principle of FMCW Radar

FMCW radar’s vital signs measurement is based on the phase term of reflected signals from the human body. Figure 1 shows a simplified block diagram of a typical FMCW radar system. The FMCW radar system consists of transmitting (TX), RF synthesizer, receiving (RX), clock, analog-to-digital converter (ADC), digital circuit, single chip microcomputer (MCU) and digital signal processor (DSP). The FMCW radar periodically transmits a chirped signal generated by a synthesizer with a frequency that increases linearly with time through a transmitting antenna, each transmitting chain has independent phase and amplitude control. The RF synthesizer creates the desired frequency, like a chirped signal that changes over time. The signal received by the receiving antenna is amplified by a low noise amplifier (LNA) and correlated with the local chirp of the mixer. Then, after low pass (LP) filtering, a sinusoidal signal containing the instantaneous frequency difference between transmitting and receiving chirps is obtained, which is called the beat signal. Finally, an analog-to-digital converter (ADC) samples the beat signal for subsequent signal processing. With this instantaneous signal, the instantaneous frequency difference can be translated into the instantaneous distance between the radar and the target.

The transmitted signal is usually a sawtooth waveform or a triangle waveform. The sawtooth waveform is adopted in this paper, and the specific mode is shown in Figure 2.

The transmitted FMCW signal can be expressed as [5]:(1)$$S_{T}\left(t\right)=Ae^{j\left(2\pi f_{c}t+\pi \frac{B}{T}t^{2}+\theta \left(t\right)\right)}\;0\;<\;t\;<\;T$$
where $f_{c}$ is the starting frequency of the chirp signal, B is bandwidth, A is the amplitude of the transmitted signal, $\theta \left(t\right)$ is phase noise, T is the width of chirp signal pulse, $\frac{B}{T}$ is the slope of the chirp signal, which represents the changing rate of the frequency.

Suppose D(t) is the motion displacement of the chest and R is the distance from the radar sensor to the human body. The distance from the chest to the radar is r (t) = D(t) + R, and the delay is $t_{d}$ = 2 r(t)/c, where c is the speed of light. The received signal can be expressed as:(2)$$S_{r}\left(t\right)=Be^{j\left(2\pi f_{c}\left(t-t_{d}\right)+\pi \frac{B}{T}{\left(t-t_{d}\right)}^{2}+\theta \left(t-t_{d}\right)\right)}$$

The RX and TX signals are mixed by two orthogonal I/Q channels, then, through low-pass filtering, an intermediate frequency (IF) signal is obtained. The IF signal contains a single tone signal with a constant frequency.

Where
(3)$$\mathrm{IF}\left(t\right)=ABe^{j\left(2\pi f_{c}t_{d}+2\pi \frac{B}{T}t_{d}t+\pi \frac{B}{T}t_{d}{}^{2}+\Delta \theta \left(t\right)\right)}$$
(4)$$\mathrm{IF}\left(t\right)=ABe^{j\left(2\pi f_{0}t+2\pi \frac{B}{T}t_{d}t+\pi \frac{B}{T}t_{d}{}^{2}+\Delta \theta \left(t\right)\right)}$$
(5)$$\mathrm{IF}\left(t\right)\approx ABe^{j\left(2\pi f_{0}t+\frac{4\pi }{\lambda }\left(D\left(t\right)+\;R\right)\right)}$$
(6)$$\psi \left(t\right)=\frac{4\pi }{\lambda }\left(D\left(t\right)+\;R\right)$$

The approximate equation in (5) is obtained by ignoring the phase square term, which is about 10^−6^ (when $\frac{B}{T}\;$ is about 10^12^ Hz/s and $t_{d}$ is about 1 ns). Residual noise phase as a delta $\Delta \theta \left(t\right)=\theta \left(t\right)-\theta \left(t-t_{d}\right)$ can be ignored in the approximation for the third time. In general, t = 1 μs, the thoracic displacement R (t) is mm grade, and can be ignored relative to R. ψ(t) varies with the change of D(t) in the range of *λ* at a fixed distance R.

After the IF signal is obtained, the fast Fourier transform (FFT) is applied to the IF beat signal. Spectral peaks correspond to different distances between subjects. The range FFT of each chirp signal represents a specific distance at different time, and the range FFT at different time represents the variation of ψ(t) in (6) with time. In order to measure the change of vital signal with time, multiple chirp signals are sent in the detection time range, which is equivalent to sampling D(t). Assuming that D(t) is sampled every Tr, it is called a frame period. Therefore, the sign information can be obtained by the phase extraction of distance FFT at continuous time.

## 3. Proposed Method

The vital signal measurement process of FMCW radar can be divided into four stages: static clutter filtering, range FFT, extraction and separation of vital signal, estimation of respiration and heartbeat rates. Static clutter filtering can filter out the non-periodic movement of the target around the test target, range FFT can detect the position of the test target. Vital signal extraction is limited to the range of this position, and respiratory and heartbeat signals can be recovered by extracting phase information in the range FFT. Finally, the respiratory rate and heart rate were obtained by frequency estimation of the recovered respiratory and heartbeat signals. In this section, we will illustrate the proposed approach from these four stages and discuss its advantages.

The main processing process is shown in Figure 3. Firstly, static clutter filtering can filter out non-periodic moving targets around the test target. Then, range FFT can detect the position of the test target and limit vital signal extraction to the range of this position. After that, the respiratory and heartbeat signal waveforms were established by extracting continuous time phase information. Finally, the respiratory rate and heart rate were obtained by frequency estimation of the recovered respiratory and heartbeat signals. In this section, we will illustrate the proposed approach from these four stages and discuss its advantages.

### 3.1. A. Static Clutter Filtering

FMCW radars emit chirped signals, and all targets in the radar radiation space scatter the radar echoes. Although FMCW radar has range detection capability, it cannot distinguish different targets within the same range cell. When there are stationary or a periodic moving targets near the monitored vibration target, the IF beat signal contains the clutter signal reflected by these targets. In particular, the non-periodic moving target that is close to the target may lead to errors in effective phase extraction, thus affecting the accuracy of the final test results. According to the principle of FMCW radar, the static target around the target has the same chirp signal distance information for each frame; therefore, as shown in Equation (7), firstly, the IF signal is accumulated, and then its average value is calculated (frames signal cycle time is a vital of integer times). The vital signal is a periodic signal so it has accumulative results of 0. The static target information contained in the IF signal can be filtered by subtracting the average value of the intermediate frequency signal in each frame, and the power of the non-periodic moving target around the target will also be greatly reduced, so as to achieve the filtering of the static target, reduce the power of the non-periodic moving target, and improve the SNR of the periodic vital signal.
(7)$$y_{\mathrm{AVG}}=\frac{\sum_{l=1}^{L}ABe^{j\left(2\pi f_{l}t+\frac{4\pi }{\lambda }\left(D_{l}\left(t\right)+{\;R}_{l}\right)\right)}}{L}$$
$y_{\mathrm{AVG}}$ is the average value of N frame echo data, L is the number of frames.

### 3.2. Phase Extraction

In fact, the phase changes measured by the IF signal are very slow, meaning that there is a long system idle time between two consecutive chirps. In non-contact vital sign monitoring applications, the frequency range of interest is about 0.1 Hz to 2 Hz (6 BPM to 120 BPM). The sampling rate of conventional FMCW radar can reach several megahertz, and the sampling length is usually on the order of several hundred. Breathing and chest displacement movements are approximately 12 mm at most, several times the FMCW radar wavelength (4 mm at 77 GHz). If we use the traditional arctangent demodulation technique to extract the phase values, the extracted phase values will exceed the phase range (−π/2, π/2), this leads to phase discontinuities, phase blurring, and thus phase hopping. The two I/Q demodulation signals can be expressed as [5,6,7]:(8)$$I\left(t\right)=A_{I}cos\left(2\pi f_{c}nT_{m}+\phi \right)+DC_{I}$$
(9)$$Q\left(t\right)=A_{Q}sin\left(2\pi f_{c}nT_{m}+\phi \right)+DC_{Q}$$
n is discrete sampling points, $T_{m}\;$ is interval time of every time sampling, *ϕ* is signal displacement information, $DC_{I}$, $DC_{Q}$ is the system offset, in order to ensure the continuity and the accuracy of phase, we adopted an extension DACM algorithm. The DACM algorithm transforms arctan function into derivative operation [5], which can be expressed as:(10)$$\frac{d}{dt}\phi \left(t\right)=\frac{d}{dt}\left[arctan\frac{Q\left(t\right)}{I\left(t\right)}\right]=\frac{I\left(t\right)Q{\left(t\right)}^{\prime }-Q\left(t\right)I{\left(t\right)}^{\prime }}{I{\left(t\right)}^{2}+Q{\left(t\right)}^{2}}$$

Finally, the integration stage is expressed in discrete form as:(11)$$\phi \left(k\right)=\overset{n}{\underset{k=2}{\sum }}\frac{I\left(k\right)\left\{Q\left(k\right)-Q\left(k-1\right)\right\}-Q\left(k\right)\left\{I\left(k\right)-I\left(k-1\right)\right\}}{I{\left(k\right)}^{2}+Q{\left(k\right)}^{2}}$$

Although the extended DACM algorithm solves the phase ambiguity problem, the heartbeat frequency is so small that it is easy to drown in the respiratory harmonic frequency and noise. Therefore, phase difference processing is performed to enhance the heartbeat signal. Phase difference is the difference between adjacent continuous phase values, namely, $\phi \left(k\right)-\phi \left(k-1\right)$. This differential phase expression can suppress phase shift and enhances the continuity of heartbeat signals.

### 3.3. Adaptive EEMD Recognition Method

After phase extraction of Tc (Tc/Tr sampling frames should cover at least one respiratory and heartbeat cycle) time, we first extracted the respiratory and heartbeat waveform, and then separated the respiratory and heartbeat waveform. In fact, since the displacement of the chest cavity is the result of the joint action of respiration and heartbeat, the phase term not only includes the components of respiration and heartbeat, but also the harmonic components of respiration [17,19,20,21]. The displacement information of vital signs $x\left(t\right)$ can be expressed as:(12)$$x\left(t\right)=x_{hr}\left(t\right)+x_{re}\left(t\right)+\overset{M}{\underset{m}{\sum }}x_{m}\left(t\right)$$

After discretization, it can be expressed as:(13)$$x\left(n\right)=x_{hr}\left(n\right)+x_{re}\left(n\right)+\overset{M}{\underset{m}{\sum }}x_{m}\left(n\right)$$
(14)$$x\left(n\right)\approx A_{hr}cos\left(2\pi f_{hr}t+\varphi_{hr}\right)+A_{re}cos\left(2\pi f_{re}t+\varphi_{re}\right)+\overset{M}{\underset{m}{\sum }}A_{m}cos\left(2\pi f_{m}t+\varphi_{m}\right)$$
$x_{hr}\left(t\right)$ is the heartbeat component, amplitude is $A_{hr}$, frequency is $f_{hr}$, phase is $\varphi_{hr}$. Breathing component is $x_{re}$(*t*), $x_{m}\left(t\right)$ is the m-th harmonic of breath. Amplitude, frequency and phase are $a_{m}$, $f_{m}$ and $\varphi_{m\;}\;\mathrm{respectively}$.

In fact, the respiration rate is about 0.1~0.5 Hz, the heartbeat rate is about 0.8~2 Hz, and the amplitude of respiratory vibration is about 10 times higher than that of heart rate. Therefore, the respiratory harmonic frequency may be close to the heartbeat frequency and have a similar or even higher amplitude [31]. Therefore, traditional methods may misjudge heartbeat signals in the frequency domain. EMD has been proved to have the potential ability to deal with this problem [27,28,29,31]. EMD can avoid the impact of respiratory harmonics on heartbeat signals to a certain extent and separate the correct heartbeat signals.

However, EMD has problems such as modal mixing and endpoint effect [27]. In order to solve these problems, Ensemble Empirical Mode Decomposition (EEMD) is proposed [32]. EEMD uses Gaussian white noise to wrap intermediate frequency signals and the dyadic filter banks of EMD. The key of EEMD is to add random white noise to the analyzed signal. However, noise in public places belongs to natural noise and is considered as a symmetrical alpha stable distribution [33,34]. Therefore, an improved EEMD method is proposed in this paper, which uses a symmetric alpha stable distribution sequence instead of a Gaussian distribution for feature extraction of vital signals.

The simulated signal is shown in Figure 4. Parameter Settings are shown in Equation (14), where $f_{re}$ = 0.32 Hz and $f_{hr}\;$= 1.54 Hz. The sampling rate was set to 20 Hz, and the phase signal was added with Gaussian white noise with a signal-to-noise ratio of 0.2.

In Figure 5a, the original time-varying phase signal is decomposed into four intrinsic mode functions (IMF). IMF2 can be identified as a respiratory component. IMF1 is the component closest to the heartbeat, but it can be seen from Figure 5b that its spectrum contains three components, which constitutes the mode mixing problem. It can be found that it is difficult to distinguish the heartbeat from the spectrum of the original signal. On the other hand, from Figure 6a, we can see that the EEMD with Gaussian white noise decomposed the original signal into 10 IMFs. The spectra of IMF3, IMF5 and original signals are shown in Figure 6b,c. The frequency corresponding to the protruding peak of IMF3 in the range of 1~2 Hz is very close to the real value of heartbeat frequency. However, the interference term still exists in the range of 0.8~1 Hz, which is identified as the third harmonic of respiration. Therefore, we propose an EEMD adaptive recognition method based on the stable distribution of alpha to improve heartbeat measurement.

The probability density function of symmetric alpha stable distribution [33] can be expressed as: (15)$$f_{a}\left(x\right)=\frac{\alpha }{\left|1-\alpha \right|\pi }x^{\frac{1}{\alpha -1}}{\int_{0}}^{\frac{\pi }{2}}v\left(\theta \right)\exp\left[-x^{\frac{\alpha }{\alpha -1}}v\left(\theta \right)\right]d\theta \;\;\;\;\left(\alpha \neq 1,x>0\right)$$
where
(16)$$v\left(\theta \right)=\frac{1}{sin{\left(\alpha \theta \right)}^{\alpha /\left(\alpha -1\right)}}cos\left[\left(\alpha -1\right)\theta \right]cos{\left(\theta \right)}^{1/\left(\alpha -1\right)}$$

We use the maximum likelihood method to estimate the value of parameter alpha.

Symmetric stable alpha distribution random variable X_noise is generated, and the original signal is wound with X_noise instead of gaussian distribution noise. The original signal is reconstructed as: (17)$$x^{i}\left(t\right)=x\left(t\right)+N_{std}X\_{\mathrm{noise}}^{i}\;\;\;\;\;t\in \left[0,\mathrm{Tr}\right]$$

The X_noise *i*, *i* = 1, 2, 3,… is an independent symmetric stable alpha distributed noise, $N_{std}$ is signal-to-noise ratio.

$x\left(t\right)$ is the original signal, The original signal is wrapped by different noises of alpha distribution, join n times, total average number of n is set. The larger n is, the smaller the reconstructed signal noise is. EMD decomposition is performed for the signals with noise addition and noise reduction, respectively, and the mean value of the decomposed two groups of IMF is calculated, and then the mean value of the N group of IMF is calculated to obtain the final IMF group component.

Suppose *x*[*n*], n = 1, 2, … *N* is the discrete form of the *x*(*t*). IMF*q* represents the *q*th IMF output obtained by the method proposed in this paper, and EMD*q*() represents the *q*th mode obtained by EMD [30].
(18)$$x_{q}^{i}\left[n\right]={\mathrm{residual}}_{q-1}\left[n\right]+EMD_{q-1}\left(X\_{\mathrm{noise}}^{i}\left[n\right]\right)$$

The *q*th IMF is:(19)$${\mathrm{IMF}}_{q}=\frac{1}{I}\overset{I}{\underset{I=1}{\sum }}EMD_{I}\left(X\_{\mathrm{noise}}^{i}\left[n\right]\right)$$

Relationship between *q*-order residual and *q* − 1 order residual as:(20)$${\mathrm{residual}}_{q}={\mathrm{residual}}_{q-1}-{\mathrm{IMF}}_{q}$$
*q*${\mathrm{residual}}_{q-1}\left[n\right]$ is the discrete form of the residual signal.

If the conditions for the end of EMD decomposition are completely followed, the original signal decomposition will get all IMF. However, we do not need to get all IMF. According to the characteristics of respiration and heartbeat signals, if we can identify the IMF components corresponding to respiration and heartbeat, we can also terminate the entire iteration without obtaining all IMF components, and the number of iterations will be reduced, thus reducing the calculation time [30]. The frequency and amplitude range of respiration and heartbeat signals are shown in Table 2.

If the sampling time window is the Tr, MaxNUM*q* is the IMF*q* maximum points, AmpMax is the maximum average, MinNUM*q* is the IMF*q* minimum point, AmpMin is minimum mean:(21)$$\sigma =\left({\mathrm{MaxNUM}}_{q}+{\mathrm{MinNUM}}_{q}\right)/2\mathrm{Tr}$$
(22)γ=AmpMax−AmpMin

Iteration termination frequency standard:(23)$$FreMin_{br}<\sigma <FreMax_{br}$$

Iteration termination amplitude criteria:(24)$$2\pi \frac{\alpha_{min}}{\lambda }<\gamma <2\pi \frac{\alpha_{max}}{\lambda }$$

$\alpha_{min}$ and $\alpha_{max}$ is minimum and maximum amplitude, respectively, breathing $FreMin_{br}$ and $FreMax_{br}$ is the minimum and maximum respiratory frequency, respectively. we can choose according to Table 1. If the inequality in Equations (23) and (24) is satisfied, the iteration is terminated; otherwise, the decomposition of the next IMF component is carried out until the resulting residual is no longer decomposable (the residual has at least two extreme values).

In order to illustrate the superiority of the proposed method, the spectral comparison of EMD, EEMD and the proposed method is presented in Figure 7. The parameter Settings are the same as those in Figure 5 and Figure 6. It can be seen from the figure that the spectra of heartbeat components obtained by using the method in this paper are purer than those obtained by using the other two methods. The prominent peaks can be clearly identified, and the corresponding frequencies of the peaks are very close to the real values. Compared with EMD, the IMF spectrum of EEMD has filtered out the respiratory and low harmonic components, indicating that the mode mixing problem can be alleviated to a certain extent. However, at the high frequency part of the spectrum, especially around 0.8 Hz, EEMD outputs a significant spectral peak, which interferes with the correct estimation of the heartbeat.

Figure 7b shows the spectrum comparison of EMD, EEMD and the heartbeat IMF obtained by the method in this paper. Compared with EMD, the IMF spectrum of EEMD filters out respiration and its low-order harmonic components, indicating that the mode mixing problem can be alleviated to a certain extent. However, in the high frequency part of the spectrum, especially around 0.9 Hz, EEMD still outputs an obvious spectrum peak. The reason for this is that the noise added to the original signal will bring residual signal after each decomposition and interfere with the subsequent modal decomposition. On the other hand, the method proposed in this paper first decomposes the noise into a series of IMF, and then adds them to the corresponding signal, reducing the noise residue after each decomposition, thus alleviating the above problems.

In order to test and compare the time of modal components obtained by EEMD proposed in this paper, EEMD and CEEMDAN under the same conditions, simulated signal decomposition was performed on a desktop computer equipped with an Intel(R) Core(TM) i5-9400 CPU (2.9 GHz) processor and 8 GB RAM(Intel Corporation, Santa Clara, CA, USA).

Table 3 shows the time used for EEMD, CEEMDAN, and the method proposed in this paper. It can be seen that, compared with EEMD, the proposed method can save about 2/5 or even half of the time. The more complex the signal is, the more the intrinsic modes are, the more the advantages of this method can be demonstrated (refer to the termination conditions of modal decomposition in this paper). In practical application, the data update interval of non-contact vital signs monitoring system is generally 2~6 s. Assuming that the rate is 20 Hz, the length of the updated sample data is consistent with the above situation, which indicates that the method in this paper has great potential in real-time processing.

### 3.4. Estimation of Heart Rate

For respiration and heartbeat signals with a finite period, the corresponding autocorrelation function will gradually become zero, and a peak value corresponding to the basic period multiple will appear in time:(25)$$R_{x}\left(m\right)=\rho \overset{k=\infty }{\underset{k=-\infty }{\sum }}x(k+m)x^{*}(k)$$where ρ is the normalization factor, k is the sampling point, m is the lag time, and ∗ is the conjugate. First, the autocorrelation function of heartbeat and respiratory signals in IMF component is calculated. Then, the time corresponding to the autocorrelation peak in the heartbeat and breathing time interval is estimated, whose reciprocal is the corresponding breathing and heart rate.

Equation (26) is the calculation method of heart rate and respiration rate:(26)$$\gamma =\frac{T_{minute}\cdot R_{\mathrm{numpeaks}\;}\cdot F_{s}}{L_{\mathrm{Size}\;}}$$
where $F_{s}$ is the sampling rate, $T_{minute}$ is measuring the duration, $L_{\mathrm{Size}\;}\;$ is breathing or heartbeat IMF component in phase, the total length of data $R_{\mathrm{numpeaks}\;}$ is breathing or heartbeat IMF component of the wave number. $T_{minute}$ = 60 s means the number of heart beats and breaths per minute.

## 4. Experimental Results

In the experiment, the 77 GHz~81 GHz Texas Instruments (TI)’ millimeter-wave sensor IWR1843 (State of Texas, US) was used in the front of the radar. The original data of the beat signal is sampled by the on-chip ADC and transmitted to the PC through the DCA1000(Texas Instruments, Dallas, TX, USA), a special data acquisition board for millimeter wave radar sensor of TI Company. After the original data is collected, MATLAB (MathWorks, Natick, MA, USA) is used for signal processing as described in Section 3. Experimental application scenarios are shown in Figure 8a. The subjects sat in front of the radar, remained stationary, and wore a smart MI3 wristband to pick up heart rates. It should be noted that, although people sit in front of the radar and remain stable, there are still noises and interference from the human body and the surrounding environment.

IWR1843 has three transmitters and four receivers, we use two transmitters TX and four receivers RX. We use time-division multiplexing (TDM) launch chirp pulse alternately. A single pair of TX/RX antennas can detect the heartbeat and breathing of an individual. We send a chirp pulse with duration Tc = 60 s alternately (In some cases, the target is far away from the radar. In order to obtain a longer detection range, two TXs can be used to transmit signals simultaneously to increase the signal gain. The idle time between pulses is 6 s, sawtooth frequency modulation slope is K = 70 MHz/μs, frame rate is 50 ms, and chirp sampling points are 256. The specific frequency chirp form is shown in Figure 8b, the ADC sampling rate is 5209 ksps. The observation time includes at least two cycles of respiratory signals. In order to better analyze the heartbeat rate and respiration rate per minute, we set the observation time T = 60 s. The specific radar parameter settings are shown in Table 4.

### 4.1. Identify Target Range

During the experiment, the human body was always in front of the radar (see Figure 8a), meaning that the regular vibrations of the target’s body were caused only by heartbeat and breathing.

As described in Section 3, a stationary object near the target and the periodic motion target will affect the accurate extraction of heart rate and breathing. In order to verify the static clutter filtering method, in this experiment, we place not only static objects but also linear moving objects around the target.

As shown in Figure 9a, before the static clutter filter processing, the radar detects moving targets and stationary objects, and it is difficult for us to distinguish the location of the tested targets. After static clutter filtering, as shown in Figure 9b, we can see that only the tested targets exist in the whole picture, which proves the effectiveness of static clutter filtering. Through this method, we can define the test range. This can avoid the influence of moving target’s displacement at the same distance from the subject to vital signal extraction and improve the accuracy of estimation of heart rate and respiration rate.

As shown in Figure 10, we can see a significant peak on the spectrum, with the target distance of 0.65 m from the radar, and the peak spread distance of about 25 cm on the spectrum. This is caused by the range resolution of the 77 GHz radar, which is the ability to distinguish two or more objects. When two objects are close enough, the radar system will no longer be able to tell them apart. We know that the radar range resolution for C/2B. As shown in Table 3, B is 4 GHz, the range resolution is 4 cm. The vibration displacement around the chest caused by breathing and heartbeat is also reflected in the radar echo.

### 4.2. Results

During the experiment, our system continuously collects data and randomly selects a frame for processing. EMD method and the method proposed in this paper are used to process the original data, and the decomposition results are shown in Figure 11 and Figure 12, respectively. As shown in Figure 11a, different modes are mixed and contribute a peak between 0.5–0.8 Hz, the amplitude of which even exceeds that of the heartbeat. In Figure 11b, it is difficult to distinguish heartbeat signals from respiratory harmonics. On the other hand, the experimental results in Figure 12 are consistent with those in Figure 7, where the respiratory and heartbeat components are successfully identified as IMF5 and IMF3, respectively. The heartbeat spectrum decomposed by this method is clearer, and the estimated frequency is 1.2 Hz, which is close to the test result of the MI3 bracelet.

In order to verify the superiority of this method, the spectral comparison of heartbeat IMF is presented in Figure 13. The results of Figure 13a–d are respectively from four randomly selected fragments of actual signals. Heartbeat component characteristics obtained by using the method in this paper are more obvious, and it is easier to determine the heartbeat frequency with this system. Although the results obtained by EEMD are much better than those obtained by EMD, there are still residual errors of respiration and its harmonic components, and the magnitude of residual errors of respiration harmonic components is larger than that of heartbeat components, which is consistent with the simulation results in Figure 6.

In order to verify the robustness of our proposed method, we conduct a long time flow experiment to verify the spectral estimation performance of our algorithm. During the experiment, the sampling rate was 20 Hz, and the stream data of 5000 frames (250 s) were analyzed with a 200-frame distance window (10 s) and a 20-frame interval time (1 s). The time-varying spectrum of heartbeat IMF is shown in Figure 14. It can be seen that in the whole-time frequency spectrum, heartbeat changes are within the range of 1.2–1.4 Hz (72–84 bpm). The reliability and robustness of the proposed method in long-term measurement are proved.

We use root mean square to evaluate the error between our proposed method and the contact test method, and we define root mean square as:(27)$$\delta =\sqrt{\frac{1}{N}\overset{N}{\underset{n=1}{\sum }}{\left(f_{EEMD}-f_{WATCH}\right)}^{2}}$$
where $f_{EEMD}$ is the heartbeat frequency measured using the method proposed in this paper, $f_{WATCH}$ is the heartbeat rate measured using MI3, N is the number of tests.

During the experiment, three adult males and three adult females were selected to test the heartbeat rate at the same radar range, and 150-s stream data were recorded to calculate the root mean square error between the method in this paper and the MI3 bracelet test results, as shown in Table 5.

The details are shown in Table 5. It can be seen that the root mean square error between the heart rate measured by the method in this paper and the results measured by the MI3 bracelet is less than 3. For the same subject, the longer the detection time, the more data frames, and the smaller the root mean square error. When the detection time is greater than or equal to 150 s, the root mean square error is basically close to 2, that is, the heart rate error tested in this paper and the heart rate error tested by MI3 bracelet is below 4 bpm. As shown in Table 6, the results of the proposed method and other methods are compared. Heart rate accuracy improved by about 5%. Experimental results show that this method has good adaptability and reliability.

## 5. Discussion

The extended DACM allows us to obtain valid heartbeat and respiration-related phase information after correcting for DC offset using the center of the circle dynamic tracking algorithm. The static clutter filtering algorithm, as shown in Figure 9, has great advantages in the extraction of periodic motion signals, reducing irregular wave phenomenon and effectively reducing noise, while retaining the time-frequency characteristics of the original signal.

EEMD algorithm is an improved algorithm of EMD, which solves the problems of modal mixing and endpoint effect in EMD algorithm. However, after the decomposition of the original EEMD algorithm, the residual noise of each IMF component will affect the effective signal. Therefore, this paper proposes an improved EEMD algorithm to extract and distinguish heartbeat and respiratory signals. Firstly, positive and negative white noise were added to the original signal, and the residual noise was reduced after decomposition. Then, according to the characteristics of ambient noise, we added a-distributed noise to the original signal. Finally, according to the characteristics of the signal itself, the number of iterations is determined adaptively, which improves the accuracy and efficiency of EEMD algorithm. In Figure 13, we use the EMD algorithm, EEMD algorithm and improved EEMD algorithm, respectively, to decompose the same signal, which is consistent with our simulation results and verifies the superiority of the improved EEMD algorithm. In order to evaluate the practical value of our algorithm, we analyzed the results obtained by contactor MI3 and the improved EEMD algorithm, as shown in Table 4. The results show that the error between the algorithm and contactor MI3 is less than 4 bpm, which has practical value, and also shows the potential and prospect of contactless detection in the future.

## 6. Conclusions

In this paper, a 77 GHz FMCW radar is used to obtain respiratory and heartbeat signals by extracting radar intermediate frequency phase information. This paper systematically introduces radar data signal processing flow and parameter configuration. Some methods are proposed to ensure the accuracy and reliability of life signal extraction, and the results are compared with those of contact equipment. A static clutter filtering method is proposed to eliminate the interference of moving and stationary objects in the target range. An adaptive EEMD method based on symmetric α stable distribution is used to extract human vital signs signals. Experimental results show that the static clutter filtering method can effectively eliminate the interference of non-periodic motion and stationary objects within the target range. The improved EEMD algorithm can effectively distinguish heartbeat and respiration components and accurately estimate heart rate with a root mean square error less than 4 bpm, which proved the feasibility and effectiveness of FMCW radar in remote vital signs monitoring.

## Figures and Tables

**Figure 1 sensors-22-06423-f001:**
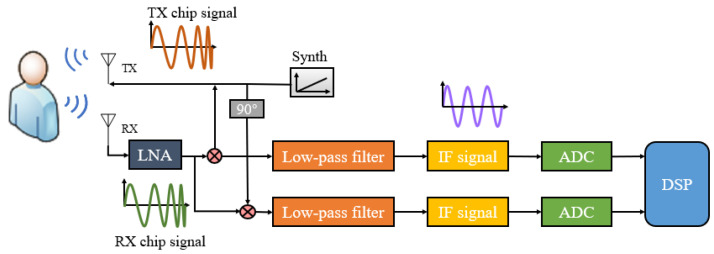
FMCW radar block diagram.

**Figure 2 sensors-22-06423-f002:**
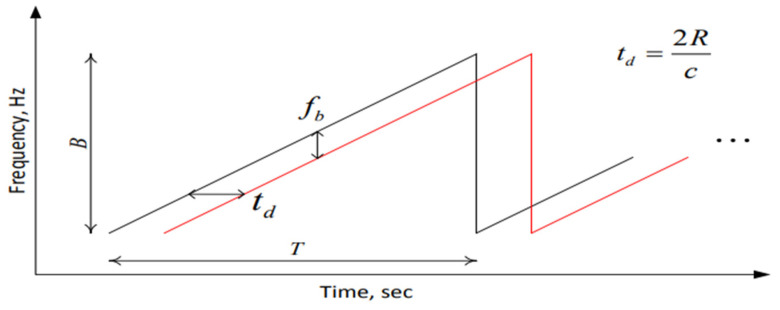
Spectrum of chirped signal frequency with time.

**Figure 3 sensors-22-06423-f003:**
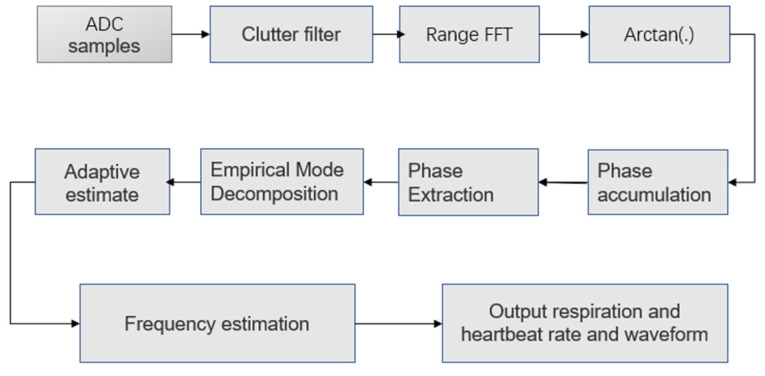
Flow chart of human heartbeat and respiration signal detection.

**Figure 4 sensors-22-06423-f004:**
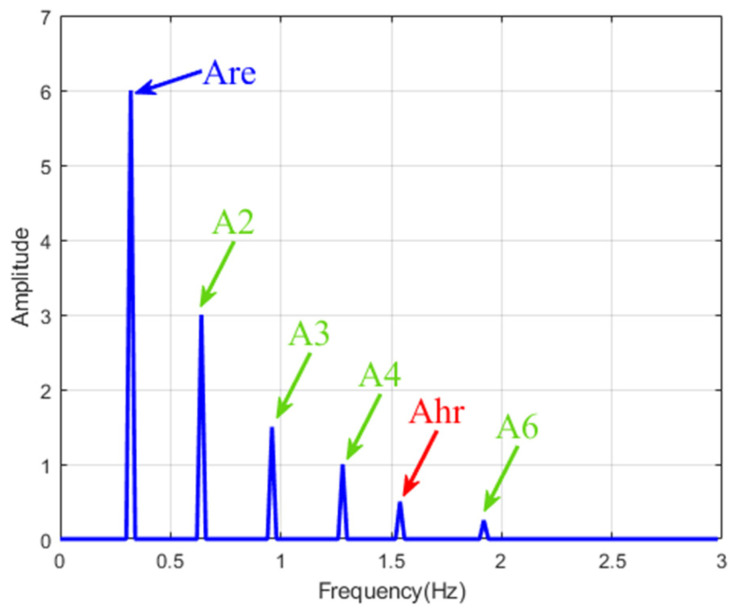
Spectrum of simulated phase signal. Respiration rate $f_{re}$ = 0.32 Hz, heartbeat rate $f_{hr}$ = 1.54 Hz. The 2nd, 3rd, 4th and 6th breath harmonics have been added. The original signal has respiratory harmonics in the range of 0.8~2.0 Hz.

**Figure 5 sensors-22-06423-f005:**
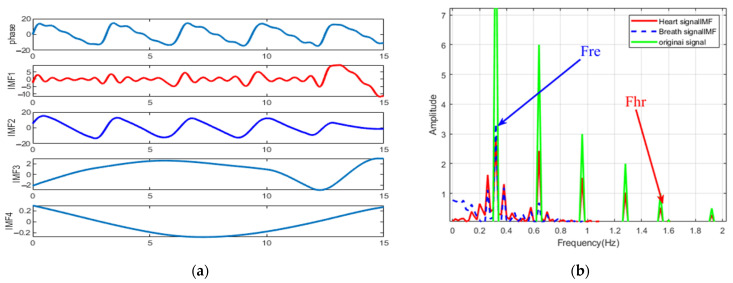
(**a**) EMD decomposition results of simulated signals, the red line is the component close to the respiratory rate, and the blue line is the component close to the respiratory rate. (**b**) Spectrums of simulated signals, respiratory and heartbeat spectrums. Respiration rate $F_{re}$ = 0.32 Hz, heartbeat rate $F_{hr}$ = 1.54 Hz. The 2nd, 3rd, 4th, and 6th breath harmonics have been added. IMF2 and IMF1 are considered to be the closest respiratory and heartbeat components, respectively. In (**b**), false peaks exist in both the original signal and the heartbeat spectrum within 0.8−2.0 Hz, making it difficult to estimate the heartbeat frequency.

**Figure 6 sensors-22-06423-f006:**
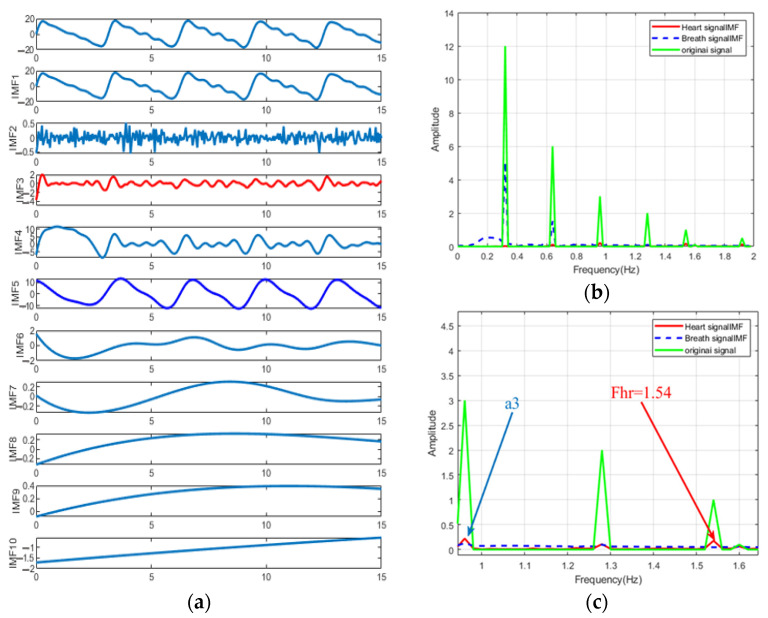
(**a**) EEMD results of simulated signals, the red line is the component close to the respiratory rate, and the blue line is the component close to the respiratory rate. (**b**) Simulated signal spectrum, respiratory spectrum and heartbeat spectrum. (**c**) Spectral details of simulated signals, respiratory and heartbeat domains. The respiratory rate and heartbeat rate are set as shown in Figure 4, that is, $F_{re}$ = 0.32 Hz and $F_{hr}$ = 1.54 Hz. The second, third, fourth and sixth breath harmonics were added, and the SNR was set to 0.2. IMF5 and IMF3 are considered to be the closest respiratory and heartbeat components, respectively. The heart rate can be estimated from the IMF3 spectrum of (**c**).

**Figure 7 sensors-22-06423-f007:**
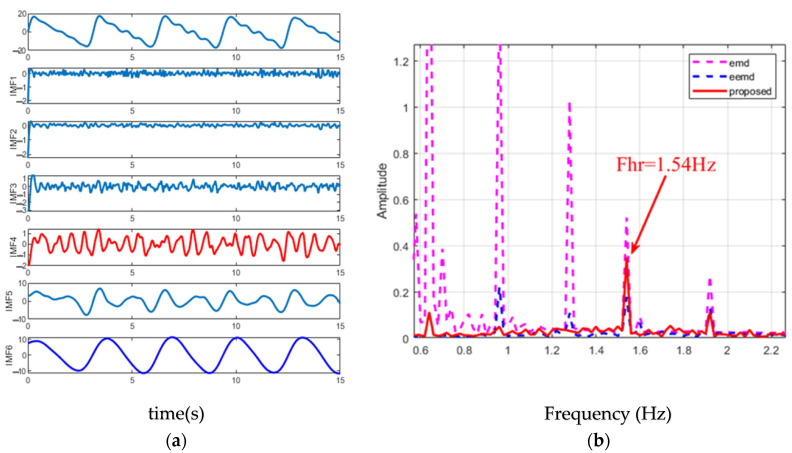
(**a**) Decomposition results of simulated signals using the improved EEMD, the red line is the component close to the respiratory rate, and the blue line is the component close to the respiratory rate. (**b**) Spectrum comparison of heartbeat IMF obtained from simulated signals using EMD, EEMD and the improved EEMD. The respiratory rate and heartbeat rate are set as shown in Figure 5, that is, $F_{re}$ = 0.32 Hz and $F_{hr}$ = 1.54 Hz. The second, third, fourth and sixth breath harmonics were added, and the SNR was set to 0.2.

**Figure 8 sensors-22-06423-f008:**
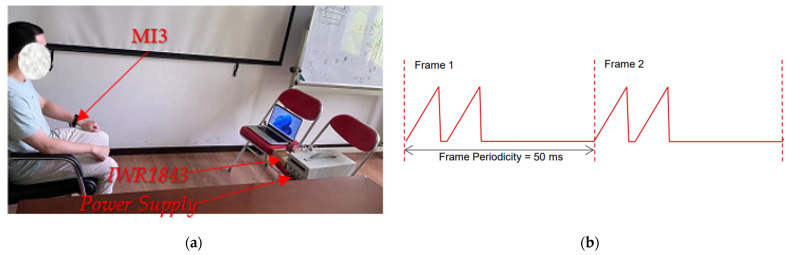
(**a**) Experimental setup, which included volunteers, FMCW radar (IWR1843) component, and a laptop. (**b**) Radar chirp parameter setting.

**Figure 9 sensors-22-06423-f009:**
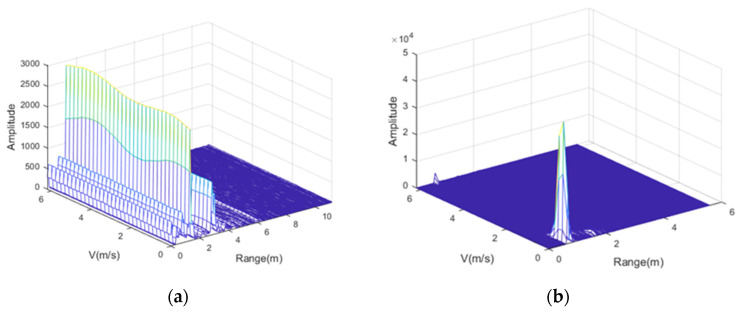
Before (**a**) and after (**b**) static clutter filtering technology for echo signal, the raised lines in the figure represent targets detected by radar, it can be seen that, before processing, the radar detects moving targets and stationary objects, and it is difficult for us to distinguish the location of the subject target. After processing, we can see that only the subject target exists in the whole picture.

**Figure 10 sensors-22-06423-f010:**
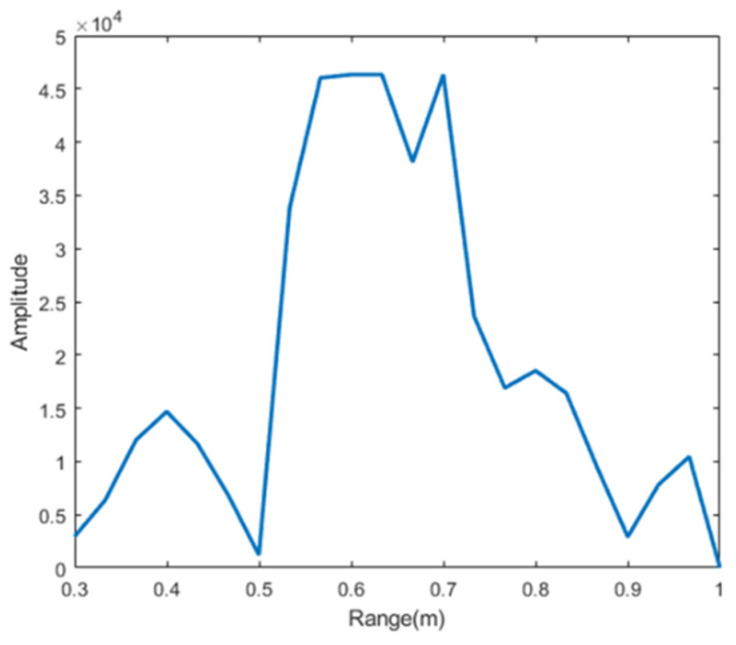
Target range, radar target distance is 0.65 m, radar range resolution is *C*/2B, range resolution is 4 cm.

**Figure 11 sensors-22-06423-f011:**
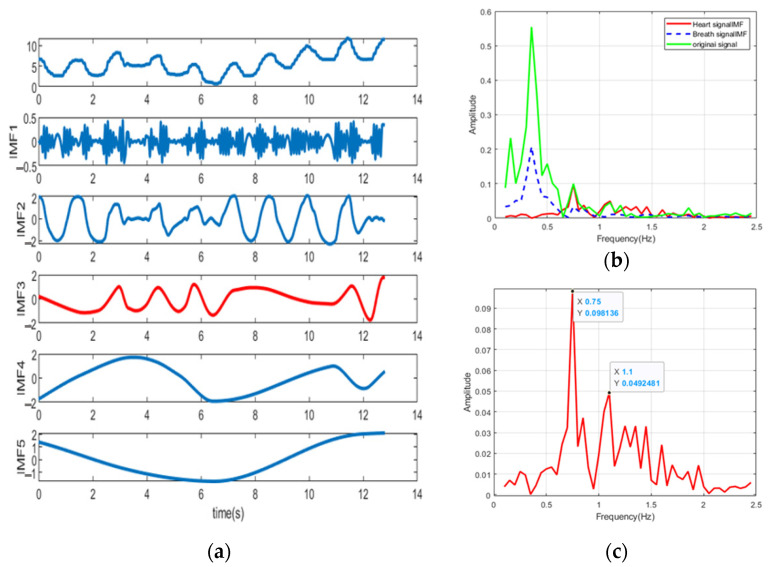
(**a**) EMD decomposition results of actual signals, the red line is the component close to the respiratory rate. (**b**) Spectrograms of actual signals, respiratory spectrograms, and (**c**) heartbeat spectrograms. IMF3 and IMF2 are considered to be the closest respiratory and heartbeat components, respectively. In (**c**), there are false peaks in the range of 0.8−2.0 Hz in both the original signal and the heartbeat spectrum, making it difficult to estimate the heartbeat frequency.

**Figure 12 sensors-22-06423-f012:**
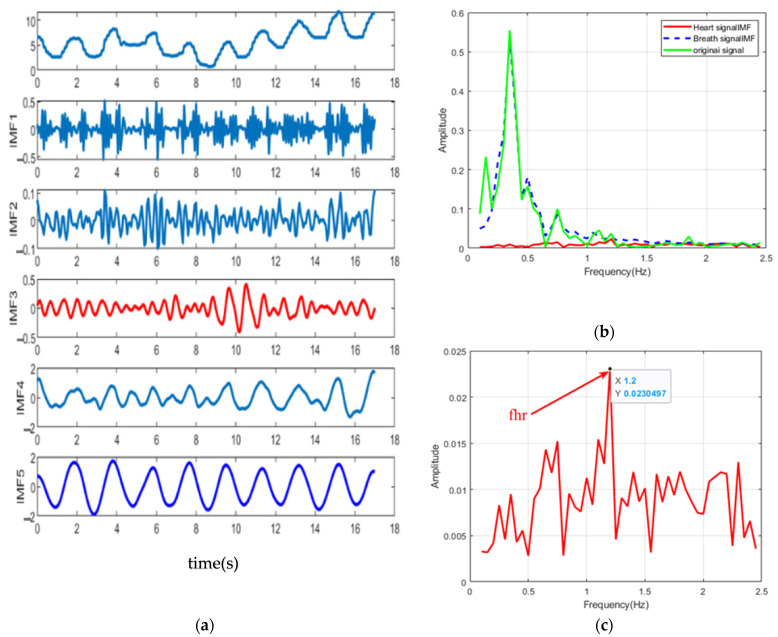
(**a**) Decomposition results of actual signals using the method proposed in this paper, the red line is the component close to the respiratory rate, and the blue line is the component close to the respiratory rate. (**b**) spectrograms of actual signals, respiratory spectrograms and (**c**) heartbeat spectrograms. IMF5 and IMF3 are considered to be the closest respiratory and heartbeat components, respectively. The heart rate of 1.2 Hz can be estimated from the IMF3 spectrum of (**c**) in the range of 0.8−2.0 Hz.

**Figure 13 sensors-22-06423-f013:**
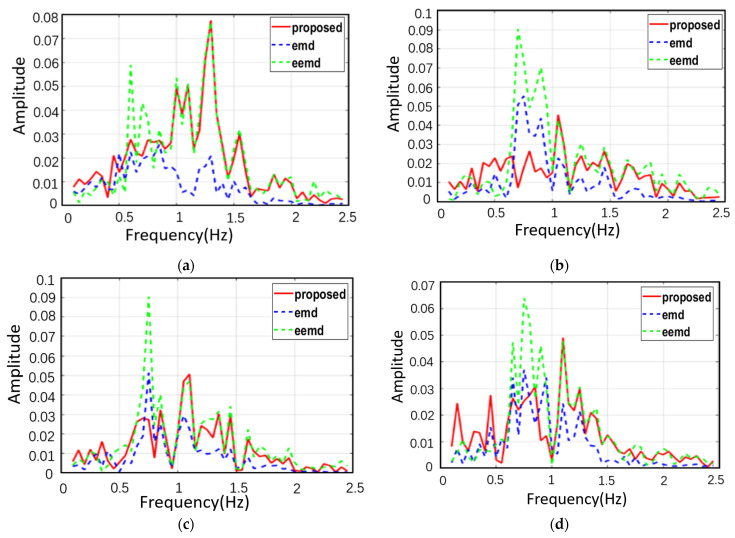
Spectrum comparison of heartbeat IMF obtained from actual signals using EMD, EEMD, and the algorithm proposed in this paper. (**a**–**d**) are the calculated results of selecting four parts from an actual signal.

**Figure 14 sensors-22-06423-f014:**
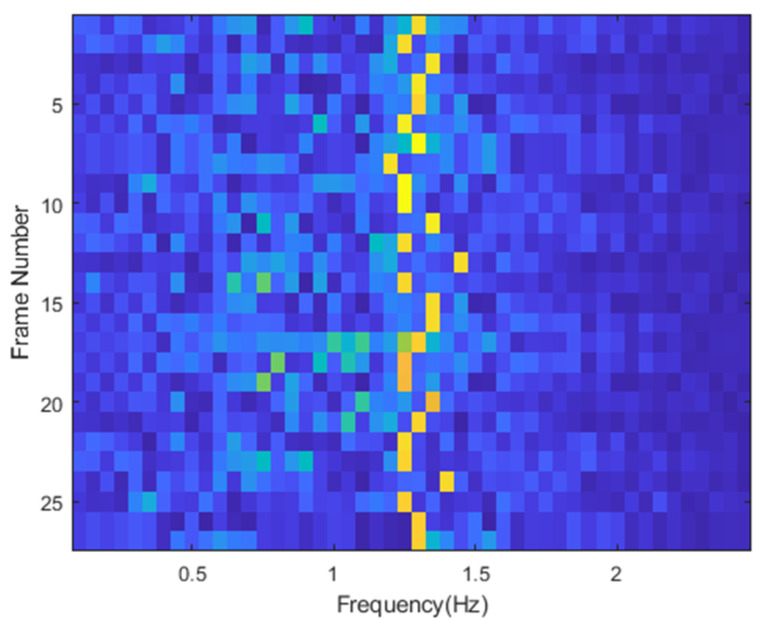
The stream processing results of the original signal, 5000 frames (250 s) of stream data, a window every 10 s, the time interval is 1 s.

**Table 1 sensors-22-06423-t001:** Vital sign monitoring with radar techniques.

Radar Technology	Frequency (GHz)	Reference Signal	Results	Processing Method	Additional Comments
FMCW [13]	24.05–24.25	Piezoelectric finger sensor	Heart rate	FFT bandpass filter	Simultaneously heart rate detection of multiple subjects. The detection range is less than 1.8 m
FMCW [1]	77–81	-	Respiratory and heart rate	FFT bandpass filter	Simultaneous vital sign detection of multiple subjects, uses MIMO. SNR > 40 dB, the phase sensitivity is <7 milli-radians which corresponds to a displacement sensitivity of ≈2 microns.
FMCW [19]	75–85	Philips MP70: ECG + CO_2_ changes	Respiratory and heart rate	FFT bandpass filter	The best result is achieved with the frontal position at 1 m distance with a median relative error of 8.09%
FMCW [4]	114–130	ECG	heartbeat waveforms, HRV, and respiratory and heart rate	two-step FFT EMD	Simultaneously vital sign detection of multiple subjects and analysis of coupling between breathing and heartbeat. The detection accuracy is 2 um. There are some problems such as mode mixing and end effect
FMCW [22]	24–24.05	belt sensor	Respiratory and position	two-step FFT ROl determination and weighting process to minimize the clutter from the debris And other objects under debris	the maximum depth of the radar system is 3.28 m behind the wall material.
FMCW [5]	77–81	Mi 3	Heartbeat rate and Breathing rate	CS-OMP RA-DWT	both heartbeat rate and respiration rate were higher than 93%
FMCW [this work]	77–81	Mi 3	Heartbeat rate and Breathing rate	DACM improved adaptive EEMD, Clutter filter	The detection range is 0.5–2.5 m, The error with contactor MI3 is less than 4 bpm, and the heartbeat accuracy is more than 95%

**Table 2 sensors-22-06423-t002:** The range of amplitude and frequency of breathing and heartbeat.

Vital Signs	Amplitude	Frequency
breath rate	1–12 mm	0.1–0.5 Hz
heart rate	0.1–0.5 mm	0.8–2 Hz

**Table 3 sensors-22-06423-t003:** Lumped mean times and calculation time.

Number of Realizations(I)	Time (s)		
	EEMD	CEEMDAN	Proposed
100	3.4573	2.9267	2.254648
200	6.065454	4.915177	3.656145
400	12.043058	8.953022	6.288670
800	24.211245	18.168884	14.939640

**Table 4 sensors-22-06423-t004:** Chirp parameter list.

**Parameter.**	B	Fc	St	Ft	Sp	Td
Value	4 GHz	77 GHz	20 Hz	5 MHz	256	57 us

**Table 5 sensors-22-06423-t005:** Root mean square error of experimental data and mi3 bracelet data.

Subjects	Root-Mean-Square		
	50 s	100 s	150 s
Male1	2.34	2.06	1.79
Male2	2.57	2.32	2.04
Male3	2.86	2.45	2.13
Female1	2.14	2.06	1.79
Female2	2.96	2.63	2.27
Female3	2.67	2.37	2.19

**Table 6 sensors-22-06423-t006:** Heartbeat rate comparison between different methods.

Subjects	Mi 3 (Reference Heartbeat Rate)	IWR1443 Radar Sensor					
		Filtering		RA-DWT		Proposed	
		FFT	Auto-Correlation	FFT	Auto-Correlation	FFT	Auto-Correlation
Male1	85	89	82	92	87	87	85
Male2	76	84	74	87	81	79	77
Male3	70	82	72	78	75	72	69
Female1	73	81	78	81	79	72	70
Female2	64	77	69	75	71	67	65
Female3	81	94	90	90	87	85	83

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
