# Peer review of "Non-Contact Detection of Vital Signs Based on Improved Adaptive EEMD Algorithm (July 2022)"

_sensors, 2022, doi:10.3390/s22176423_

Round 1

Reviewer 1 Report

The topic presented in this contribution is potentially interesting for the present journal.

However, the text needs a grammatical revision. Some technical contents need to be clarified. The novelty of the paper should be clarified and highlighted.

There are a lot of typos throughout the text.

Detailed comments:

Line 37 pag. 1: “However, continuous wave Doppler radar lacks modulation spectrum information, so it is not easy to distinguish background clutter and complete multi-target detection”. You were speaking about FMCW, then you wrote Doppler but the lack of modulation seems referred to Doppler. In the next sentences again FMCW and Doppler mixed. At line 43 another typo. I suggest reorganizing the period. You can refer to doi.org/10.1063/5.0044673 for more information concerning FMCW radars.

Also the statement: “Doppler radar cannot reduce other targets and environmental interference, resulting in low accuracy of vital signs detection” is not clear. Indeed, Doppler radars are considered reliable for vital sign detection.

Pag 2, line 53: “In addition, as far as we know, the current FMCW processing scheme for vital signs monitoring is mainly based on a single chirp and data from a distance bin, which Cannot transmit chirp signal continuously. Heart rate and respiration rate cannot be measured in real time.”. Again there are typos but above all the phase-based detection of vital sign in FMCW radars is based on multiple chip. Again, you should re-write all these theoretical considerations. You can refer to the following reference:

Adiprabowo, T.; Lin, D.-B.; Wang, T.-H.; Purnomo, A.T.; Pramudita, A.A. Human Vital Signs Detection: A Concurrent Detection Approach. Appl. Sci. 2022, 12, 1077. https://doi.org/10.3390/app12031077.

There area a lot of contributions concerning ensemble empirical Mode decomposition or vital sign detection with clutter identification. You should highlight what are the novel elements of your contribution.

Another typo: “fre” at line 204.

Please can you clarify how did you extend the range by means of two transmitters? (line 347 pag. 11).

Please can you clarify the meaning of “Figure10. Target range, radar target distance is 0.65 m, at the same time extension on the frequency spectrum peak around 25 cm, radar range resolution is ?/2B, range resolution is 4 cm, 25 cm is the chest width of the subject.”. Only relevant information should be placed in the figure description.

Reviewer 2 Report

In this manuscript, the authors proposed a static clutter filtering method and an EEMD algorithm based on the characteristics of noise distribution. In my opinion, this manuscript is interesting to the readers of Sensors. The topic is very important in this field. This work is novel and original. The authors have solid background in this field. Therefore, the referee recommends it to be published after the following revisions:
1. The English should be polished by a native speaker.
2. The quality of Fig. 2 should be improved.

3. Please combine Figs. 6(a), 6(b) and 6(c) in a single page.

4. Please modify the quality of Fig. 7(b). The word “frequency” is cut off.

5. Please cite more recent works (2020-2022).

In general, this work seems to be very interesting. The referee would like to see the revision if possible.

Reviewer 3 Report

The system proposal is indeed useful for monitoring vital sign of patients.  The paper is well written although there are several issues that must be attended.

1.       The detections capacity of respiratory and heartbeat signal must be discussed with more detail.

2.       It is necessary to compare the performance of proposed system with other previously developed systems, at least in a qualitative point of view.

3.       It is necessary to comment about the accuracy detection of developed system.

Round 2

Reviewer 1 Report

The authors addressed all my concerns.

Reviewer 2 Report

This review has been carefully considered. The author answered the questions raised carefully. All the questions have been corrected and reasonably explained. I think that the paper can be accepted for publication .

Reviewer 3 Report

The authors attended the reviewer comments.